# Trial-by-trial predictions of subjective time from human brain activity

**Maxine T. Sherman**[1,2,3]*, **Zafeirios Fountas**[4], **Anil K. Seth**[1,2,5], **Warrick Roseboom**[1,2,6]*

**1** Sackler Centre for Consciousness Science, University of Sussex, Brighton, United Kingdom, **2** Department of Informatics, University of Sussex, Brighton, United Kingdom, **3** Brighton and Sussex Medical School, University of Sussex, Brighton, United Kingdom, **4** Wellcome Centre for Human Neuroimaging, University College London, London, United Kingdom, **5** Canadian Institute for Advanced Research, Program on Brain, Mind, and Consciousness, Toronto, Canada, **6** School of Psychology, University of Sussex, Brighton, United Kingdom

* maxinesherman@gmail.com (MTS); wjroseboom@gmail.com (WR)

**Data Availability Statement:** Data and materials availability: The pre-registration document, along with all data and analysis code are freely available to download at osf.io/2zqfu.

## Abstract

Human experience of time exhibits systematic, context-dependent deviations from clock time; for example, time is experienced differently at work than on holiday. Here we test the proposal that differences from clock time in subjective experience of time arise because time estimates are constructed by accumulating the same quantity that guides perception: salient events. Healthy human participants watched naturalistic, silent videos of up to 24 seconds in duration and estimated their duration while fMRI was acquired. We were able to reconstruct trial-by-trial biases in participants' duration reports, which reflect subjective experience of duration, purely from salient events in their visual cortex BOLD activity. By contrast, salient events in neither of two control regions–auditory and somatosensory cortex–were predictive of duration biases. These results held despite being able to (trivially) predict clock time from all three brain areas. Our results reveal that the information arising during perceptual processing of a dynamic environment provides a sufficient basis for reconstructing human subjective time duration.

## Author summary

Our perception of time isn't like a clock; it varies depending on other aspects of experience, such as what we see and hear in that moment. Previous studies have shown that differences in simple features, such as an image being larger or smaller, or brighter or dimmer, can change how we perceive time for those experiences. But in everyday life, the properties of these simple features can change frequently, presenting a challenge to understanding real-world time perception based on simple lab experiments. To overcome this problem, we developed a computational model of human time perception based on tracking changes in neural activity across brain regions involved in sensory processing (using non-invasive brain imaging). By measuring changes in brain activity patterns across these regions, our approach accommodates the different and changing feature combinations present in natural scenarios, such as walking on a busy street. Our model reproduces

**Funding:** This work was supported by the European Union Future and Emerging Technologies grant (GA:641100) TIMESTORM – Mind and Time: Investigation of the Temporal Traits of Human-Machine Convergence (WR, AKS), and by the Dr Mortimer and Theresa Sackler Foundation (MTS and AKS). AKS is also grateful to the Canadian Institute for Advanced Research (CIFAR) Program in Brain, Mind, and Consciousness. The funders played no role in the study design, data collection and analysis, decision to publish, or preparation of the manuscript.

**Competing interests:** The authors have declared that no competing interests exist.

people's duration reports for natural videos (up to almost half a minute long) and, most importantly, predicts whether a person reports a scene as relatively shorter or longer–the *biases* in time perception that reflect how natural experience of time deviates from clock time.

## Introduction

How do we perceive time in the scale of seconds? We know that experience of time is characterized by distortions from veridical "clock time" [1]. These distortions are reflected in common expressions like "*time flies when you're having fun*" or "*a watched pot never boils*". That our experience of time varies so strongly in different situations illustrates that duration perception is influenced by the content of sensory experiences. This is true for low level stimulus properties, such as motion speed or rate of change [2–4], mid-level properties like complexity of task [5], and more complex natural scene properties such as scene type (e.g. walking around a busy city, the green countryside, or sitting in a quiet office; [6,7]). It is also well-established that perception of time differs if attending to time or not [7,8]. That disruptions in time experience (i) arise across these different levels of stimulus complexity and (ii) are based on internal properties of the perceiver (such as what they are attending to) suggests that an approach is required that considers what is common across the hierarchy of perceptual processing, not just at a single level. By identifying a measure that captures what is common across these features and levels of complexity and basing a model of subjective duration on it, our goal is to accommodate and bridge the many previously established relationships between content and time. Further, while many studies have attempted to find a mapping (usually in the form of a correlation or similar analyses) between single, simple stimulus features and time perception (e.g. speed or temporal frequency [2–5]), natural scenes contain varying proportions of any single feature and these proportions will vary over time. Therefore, modelling subjective time perception on the scale of natural stimulation will require an approach that jointly considers the contributions of these different features.

We recently proposed [6,7,9] that the common currency of time perception across processing hierarchies is change. In principle, this is not an entirely new idea, with similar notions having been suggested in philosophy [10] and in the roots of cognitive psychology of time [5,11,12]. However, in this more recent proposal, there is a strong distinction in that change is not considered only as a function of changes in the physical nature of the stimulus being presented to the observer, but rather change is considered in terms of how the perceptual processing hierarchy of the observer responds to the stimulation.

The advantage of taking an observer-, rather than experimenter-oriented approach is that we can accommodate the critical distortions that distinguish subjective duration from veridical 'clock' time. The potential of such an approach was previously demonstrated by Roseboom and colleagues [6], who used a deep convolutional neural network that had been trained to classify different images as a proxy for human visual processing. In that study, it was reported that simply by accumulating salient changes detected in network activity across network layers it was possible to replicate biases in human reports of duration for the same naturalistic videos. This finding supported the proposal that activity in human perceptual processing networks in response to natural stimulation could provide a sufficient basis for human time perception.

The neural network used in the previous study provided a reasonable stand in for human visual processing, demonstrating at least some of the useful functional properties of human visual processing hierarchy, such as its hierarchical arrangement, specialization of layers for

different features, and increasing complexity of representations at higher layers [13,14]. There is ongoing debate about the degree and nature of similarities between such networks and biological vision, though relationships between classification performance and degree of representational similarity with primate visual processing can be found [15]. Nonetheless, a full assessment of the above proposal requires neural as well as behavioral evidence from human participants. Here, we put this proposal to a considerably stronger test, using a pre-registered, model-based analysis of human functional neuroimaging (BOLD), collected while participants estimated the duration of silent videos. In support of our proposal, we found that the model-based analysis could produce trial-by-trial predictions of participants' subjective duration estimates based on the dynamics of their multi-layer visual cortex BOLD while they watched silent videos. Control models applied to auditory or somatosensory cortex could produce reasonable estimates of clock time, but these models did not predict participants' subjective trial-by-trial biases. Our model is, to our knowledge, the first that can predict trial-by-trial biases in subjective duration purely from measured human brain activity during ongoing naturalistic stimulation.

## Results

Using functional magnetic resonance imaging (fMRI) and a pre-registered preprocessing and model-based analysis pipeline (osf.io/ce9tp), we measured BOLD activation while 40 human participants watched silent videos of natural scenes (8–24 seconds each) and made duration judgements on a visual analogue scale ranging from 0 to 40 seconds (see Fig 1A). Half of the videos depicted busy city scenes with many things happening (e.g. cars or buses going past, many people on a busy street), and the other half, office scenes with relatively few (e.g. occasionally someone would leave or enter the office). While city versus office differed broadly in how busy the content was by design, there was also substantial natural variation within the scene types.

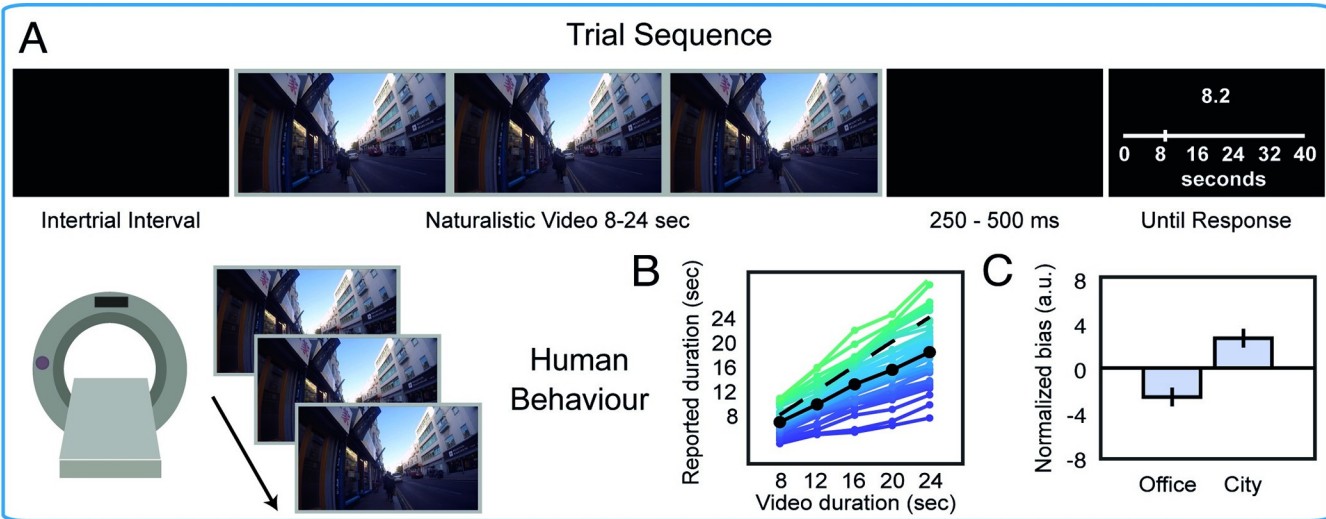

**Fig 1. Trial sequence and human behavioral results. (A)** Participants viewed naturalistic videos (8–24 seconds in duration, 1 video per trial) of walking around a busy city or sitting in a quiet office while in the MRI scanner and reported the duration using a visual analogue scale. **(B)** Participant-wise relationship between report and duration (colored lines), mean relationship (solid black line), and the line of unity (dashed line). **(C)** Relative under-/over-estimation of duration by human participants for office/city videos. Error bars represent +/- within-subject SEM.

We reasoned that if subjective time is constructed from the accumulation of salient changes in the perceptual processing hierarchy, then we should be able to predict human over- or under-estimates of time from salient changes in visual cortex dynamics. We concentrated on over- or under-estimates of time rather than correspondence with clock time because the latter would be a particularly weak (and flawed) test of our hypothesis: the accumulation of *any* positive quantity–be it salient events in visual cortex dynamics while viewing a video or number of births in Brazil while reading this paper–will increase with and therefore positively correlate with elapsed time. Our hypothesis refers to visual cortex because our stimuli are silent videos. Put simply, we predict a correlation between human behavioral biases on the one hand, and biases constructed from salient changes in visual cortex dynamics on the other (i.e. model-predicted biases). At a coarse scale, according to previous neural network based results for this video set [6], city scenes should generally produce more salient changes in perceptual processing and be overestimated relative to office scenes.

For each trial, reported durations (in seconds) were transformed into our main behavioral measure: participants' bias towards under- or over-reporting of duration. This was quantified using a (pre-registered) normalized bias measure that we have used previously [6]. Normalized bias is a trial-by-trial measure that simply quantifies the percentage difference between each duration report and the participant's mean report for the video duration category. In this way, it tells us whether any given report was high or low as compared to typical reports by the participant made under comparable conditions. For each of the $k$ trials in which the presented video duration was $t$, the bias on trial $k$ was the report on trial $k$, minus the mean report for that duration, divided by the mean report for that duration:

$$bias_{t_k} = \frac{x_{t_k} - \bar{x}_t}{\bar{x}_t} \tag{1}$$

Positive/negative values mean that individual duration reports were over-/under-estimated relative to the participant's mean for a given presented video duration. For example, on a given trial that is physically 8 seconds long, if normalized bias is -0.5, then the report on that trial was 50% less than the average for 8 second videos. If normalized bias is 0 then the report was equal to the mean, and if it is 0.5 then the report was 50% greater than the mean. Note that normalized bias takes highly similar values when calculated using the median instead of mean (for all participants r > 0.9). Because normalized bias does not take the true video duration as an input it is a bias (not accuracy) measure that reflects participant specific response patterns that are independent of clock time. This measure was then carried forward for subsequent analysis.

All inferential tests reported were preregistered unless specified otherwise.

## Behavioral reports are biased by scene type

Participants could estimate duration well, as indicated by a strong correlation between presented (veridical) and reported (subjective) durations for each subject both when computed trial-by-trial ($\bar{\rho} = 0.76 \pm 0.02$), and when averaged within duration categories ($\bar{\rho} = 0.96$, Fig 1B). As predicted, durations of city scenes were relatively over-estimated and office scenes under-estimated, M±SE$_{diff}$ = 5.23 ± 1.69 (normalized bias, %), 95% CI [1.81, 8.65], $t_{39} = 3.09$, $p = 0.004$, d = 0.50, BF$_{H(0,10.5)} = 33.8$, confirming that natural scenes containing a higher density of salient events do indeed feel longer (Fig 1C. Our pre-registered prior for the Bayes factor came from the difference of 10.5 found in [6], see Fig 3G there). Note that this result shows that the *amount* of experienced time was lower for office videos, not necessarily that time *passed faster* for office videos.

### Estimates generated by an artificial network model are biased by scene type

It has previously been shown that estimates of duration based on changes in activation across the hierarchy of an artificial image classification network can replicate human-like biases in duration reports for naturalistic stimuli [6,7]. Following from this work, we tested whether the effect of scene type for the stimuli used in our experiment and shown by our participants (Fig 1C) could be reproduced by this same artificial perceptual classification network approach.

As in the previous study [6], we fed the same video clips that participants had viewed to a pre-trained (i.e. not trained on our stimulus set) hierarchical image classification network, AlexNet [16]. For each network node, we computed frame-to-frame Euclidean distances in network activity. Then, separately for each network layer, each distance–or change in activation–was categorized as salient or not. Note that a salient change is not necessarily *psychologically* salient, nor even a salient change *in the environment*; it is simply a relatively extreme change in dynamics. Saliency categorization was achieved using an attention threshold with exponential decay that simply determined whether the change in node activation (the Euclidean distance) was sufficiently large to be deemed salient (see Methods). By decaying from the starting point to its minimum point, the threshold can adapt to local periods with few extreme changes. Following models of episodic memory [17] moments of threshold crossing are hereafter called 'salient events' (see also Discussion section *"Surprise", time perception, and episodic memory* and [7]).

Salient events were accumulated at each layer and converted to estimates of duration in seconds via multiple linear regression, by mapping the number of accumulated salient events to the *presented (clock time)*, not *reported* durations. This placed network predictions and human reports onto the same scale (seconds), and means that the model is attempting to reproduce clock time duration based on the input, rather than the more trivial task of training the model to directly reproduce human estimates. Therefore, any human-like biases in estimates can be attributed to the behavior of the network in response to the input stimuli, and not simply to the model being trained to specifically reproduce human biases.

As was the case with human behavior, and as expected, the artificial classification network-based model produced duration reports that were significantly correlated with the video duration $\rho(2329) = 0.73$, $p < 0.001$ (Fig 2A). Like our human participants, the model underestimated longer durations. As explained in Roseboom et al [6], this 'regression to the mean' effect is likely a product of mapping "sensation" (here, the accumulated salient events) onto a scale for report (here, seconds).

More importantly, the model reproduced the pattern of subjective biases seen in human participants, despite being trained on presented video duration (Fig 2B). Specifically, model-produced estimates differed as a function of video type: estimation bias was greater (i.e. reports relatively over-estimated) for busy city scenes than for office scenes, $M\pm SE_{office} = -5.00 \pm 0.66$, $M\pm SE_{city} = 4.99 \pm 0.55$, 95%CI = [8.31, 11.67], $t_{2329} = 11.65$, $p < 0.001$, d = 0.48. These results demonstrate that simply tracking the dynamics of a network trained for perceptual classification while it is exposed to natural scenes can produce the basis for human-like estimates of duration.

### Reconstructing human-like duration reports from visual cortex BOLD

Here we put our proposal to the key test. Our proposal is that tracking changes in perceptual processing in the modality-specific human sensory hierarchy is sufficient to predict human trial-by-trial biases in subjective duration. Perceptual processing of visual scenes is achieved primarily in visual cortex, so to test our proposal we asked whether we could reproduce participants' estimation biases from salient events in visual cortex BOLD. In other words, instead of

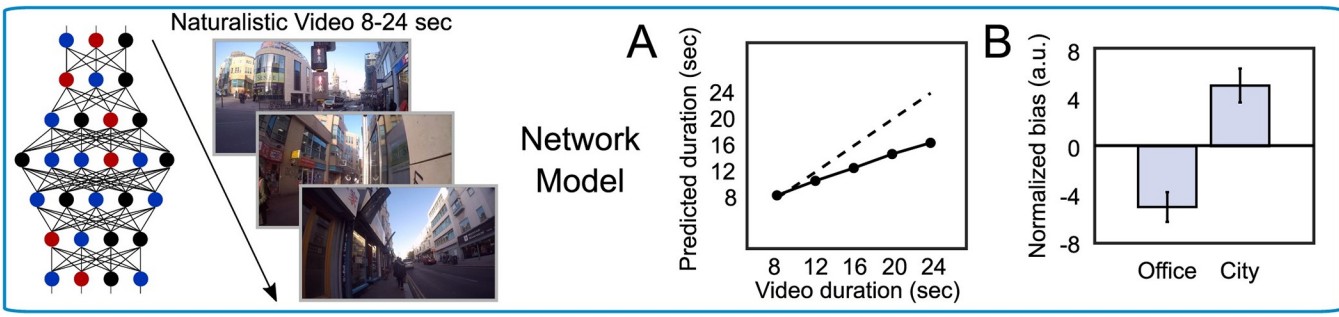

**Fig 2. Artificial network model results.** The same naturalistic videos (8–24 seconds in duration) that human participants viewed were input to an image classification network-based model to generate estimates of duration. **(A)** Relationship between presented and model-predicted video durations for this model, trained on accumulated salient events in video frames (solid line). The dashed line is the line of unity. **(B)** Relative under-/over- estimation of duration for office/city scenes for this model. Error bars represent SEM.

accumulating salient events in visual *stimulation*, we accumulated salient events in BOLD responses *to* that stimulation.

Coarse-level regional differences in BOLD were seen for both office versus city videos, and for videos (from either category) for which reports were strongly biased (GLM results, see S1 Fig and S4 Table). However, these results do not tell us about the relationship between duration biases and salient events in BOLD dynamics. If we can predict trial-by-trial subjective duration only from participants' BOLD responses in visual cortex (and not in other control regions), then we will have shown that the basis for human subjective duration judgements (when viewing natural visual scenes) can be constructed from brain activity associated with perceptual processing.

To do this, we defined a three-layer visual hierarchy *a priori* predicted to be involved in processing of the silent videos (see Fig 3 and S1 Table). We selected regions such that lower layers

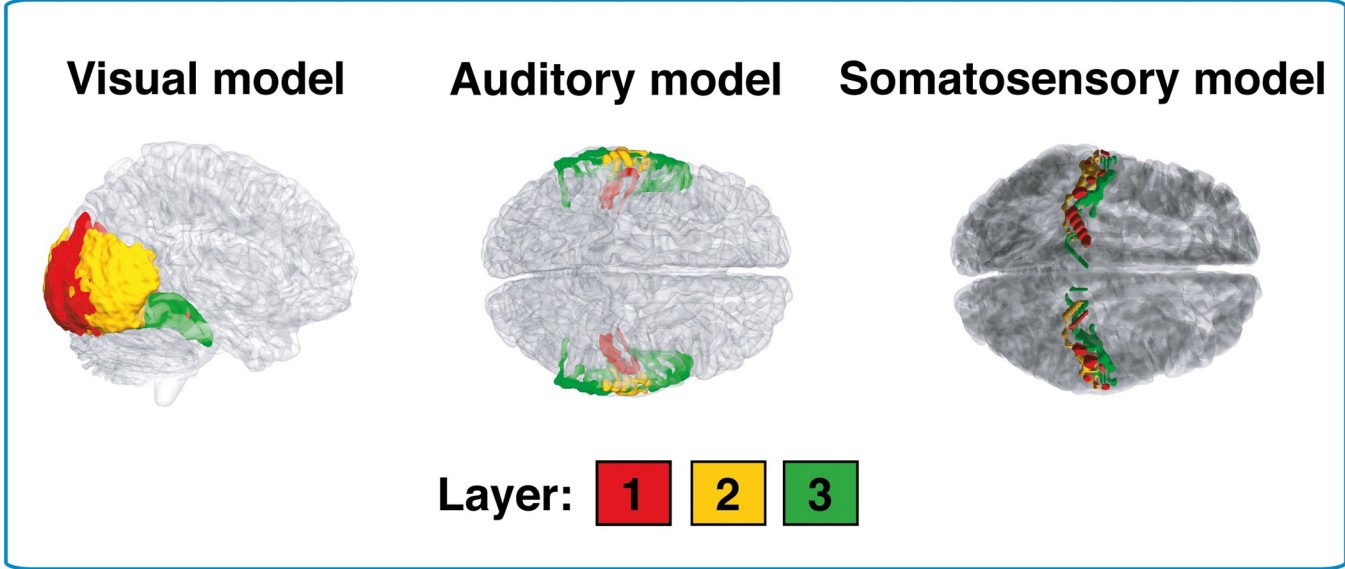

**Fig 3. Perceptual hierarchies used for fMRI-based model analysis.** Three different three-layer perceptual hierarchies were defined: a visual hierarchy, an auditory hierarchy and a somatosensory hierarchy. The visual hierarchy constitutes our model of interest, while the auditory and somatosensory hierarchies constitute control models. The regions chosen for layers 1, 2 and 3 are colored in red, yellow and green respectively. Precise details of the regions are specified in S1 Table.

reflect the processing of low-level features (e.g. edge detection in primary visual cortex; V1), and higher layers, object-related processing (e.g. lateral occipital cortex; LOC). For control analyses, analogous hierarchies were built for auditory cortex and somatosensory cortex (see S1 Table). Because the stimuli we used were silent videos, we predicted that only the model using the visual cortex hierarchy should reconstruct subjective human duration reports from accumulated salient events (see pre-registration at osf.io/ce9tp).

We ran our key analysis in two ways: one was confirmatory (i.e. pre-registered) and one was exploratory (i.e. not pre-registered). The analysis pipeline is illustrated in Fig 4. In both analyses, for each participant voxel-wise patterns of BOLD were extracted from each TR (slice, or time point) in each hierarchical layer. Voxel-wise changes between each TR were calculated and then summed over all voxels in the layer, resulting in one value per TR and layer. These 'change' values were standardized (z-scored) within-participant and compared to a criterion with exponential decay (and pre-registered parameters) to classify the change value as a salient event or not, giving us the number of salient events 'detected' by each layer for each video. Just as salient visual events would be expected to correspond to large changes in (layer-wise) visual cortical activity, salient auditory events would be expected to correspond to large changes in auditory cortex dynamics, and may be trigged by, for example, hearing (or possibly imagining) a loud sound, and similarly for somatosensory cortex. Note that psychologically salient events need not map to salient events in BOLD; see Discussion section *Predictive processing as a potential mechanistic basis for time perception*.

For the pre-registered analysis, change was quantified as Euclidean distance (as for the artificial network model), i.e.

$$\Delta_{TR} = \sum_v |X_{TR,v} - X_{TR-1,v}| \qquad (2)$$

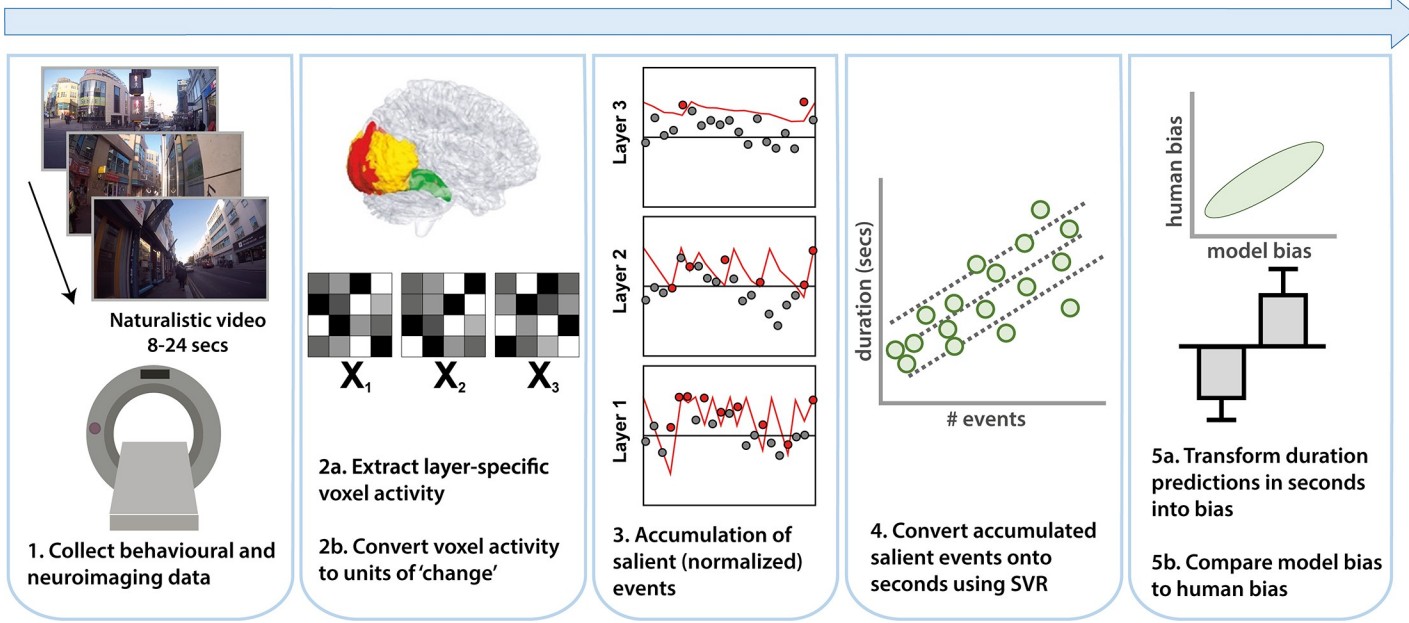

**Fig 4. Schematic of modelling analysis pipeline.** (1) Following data collection, (2a) voxel-wise BOLD amplitude was extracted and (2b) TR-by-TR (i.e. time point-by-time point, TR = repetition time) changes (Euclidean distance or signed difference) computed. The example given here is for the visual hierarchy, where each shaded matrix $X_k$ illustratively represents voxel-wise BOLD amplitudes (shaded squares) at each slice. The same process was conducted for the auditory and somatosensory hierarchies (see Fig 3 and S1 Table for different hierarchies). (3) Total change in the layer/ROI at each TR was compared to a dynamic attention threshold (red line) that categorized events as salient (red dots) or not (grey dots). The black line represents 0. An event was classified as salient if it took an equal or higher value than the threshold. (4) Accumulated salient events were regressed onto seconds, (5a) predictions from the model were converted into normalized bias (5b) and compared across condition and with human behavior.

where $X_{TR,v}$ is activation in voxel $v$ at slice $TR$. Note that (2) is mathematically equivalent to the L1 norm of the difference between BOLD at two successive TRs. However, we refer to (2) as Euclidean distances, summed over all voxels because we are proposing that the key computation here is $|X_{TR,v}-X_{TR-1,v}|$ and not $X_{TR,v}-X_{TR-1,v}$.

For the exploratory analysis, we tested an alternative algorithm for quantifying change:

$$\Delta'_{TR} = \sum\nolimits_{v}(X_{TR,v} - X_{TR-1,v}) \tag{3}$$

which we refer to as the signed difference. The attention threshold used in this analysis was the same as that pre-registered for the confirmatory analysis. We chose this measure because, at least in sensory cortices, BOLD may already reflect perceptual changes [18], potentially in the form of "prediction errors". Therefore, while the model using Euclidean distance (Eq 2) as the change metric assumes that BOLD relates *directly* to neural activity (conceptually the same as "activation" of nodes in the artificial classification network), signed difference (Eq 3) is more closely aligned with the idea that BOLD (in early sensory networks in this case) indicates (computational) *prediction error*. Euclidean distance can only be positive valued (0 or above), while the signed difference can be positive or negative in value (above or below 0).

We then used support vector regression with 10-fold cross-validation to predict the presented (i.e., clock time, not subjective/reported) video durations from accumulated salient events in layers 1, 2 and 3 for each perceptual hierarchy. This converted the accumulated salient events in the three layers to a model-predicted duration "report" in seconds so that they could be compared with human reports that were also made in seconds. Accordingly, the predicted durations in seconds by themselves are not the primary target of investigation. However, this regression step that involves an external metric of time was only necessary for directly comparing model output with human reports made in these units–as can be seen in Fig 5, accumulated salient events (defined according to Eq 3) in visual cortex already distinguish between video type prior to transformation into the units of this external metric of seconds.

Finally, biases in model predictions were compared to participants' duration estimation biases. For our pre-registered analysis, we pooled human participants' behavioral data together to create one 'super-subject', by standardizing behavioral duration reports within-participant and re-computing normalized bias using the combined behavioral dataset. For the exploratory analysis, human estimation bias was computed separately for each of the 40 participants because pooling participants' data reduced the effect of video type on (human) normalized bias (see S2B Fig). Model predictions were generated from pooled accumulated changes, regardless of whether the behavioral data were pooled or not. We did this because the use of long stimulus presentation intervals (up to 24 seconds) meant that for each participant we could only obtain relatively few trials—insufficient to generate brain-based model predictions on a purely participant-by-participant basis.

## Using Euclidean distance, estimation bias but not effects of scene type can be reconstructed from visual cortex BOLD

Following (pre-registered) pooling into a super-subject, the z-scored reports remained correlated with video durations (S2A Fig) but did not significantly discriminate between office and city scenes (S2B Fig). The presented (clock time) video duration could be predicted from accumulated salient events in all three models (visual, auditory, and somatosensory) to a similar degree (10-fold cross validation, $\bar{r}_{visual} = 0.93$, $\bar{r}_{auditory} = 0.95$, $\bar{r}_{somatosensory} = 0.94$, S2C–S2E Fig). These results show that all models could reproduce clock time–the physical duration of the presented video—and therefore that the support vector regression model successfully

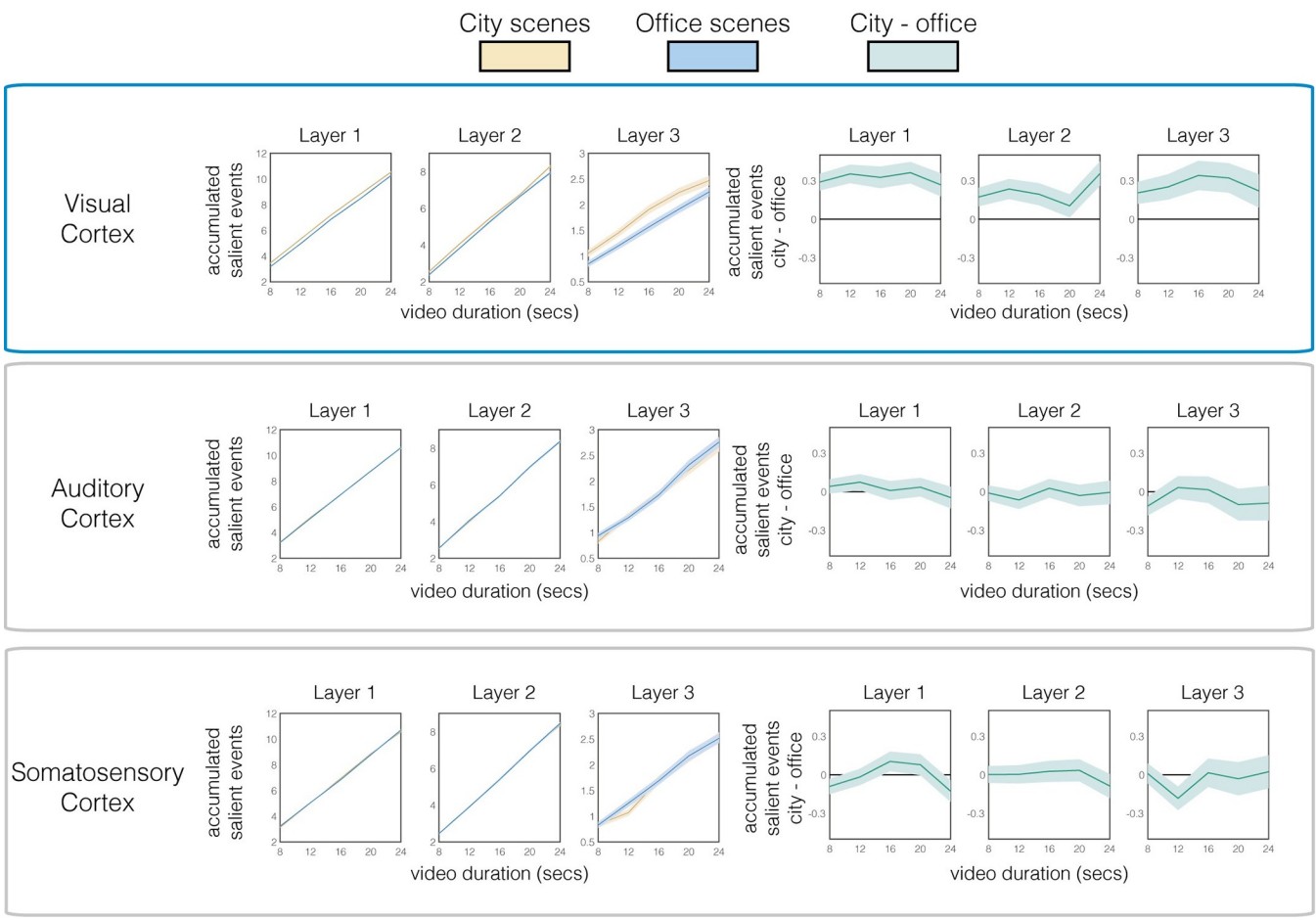

**Fig 5. Accumulated salient events over video types, perceptual hierarchies (rows) and layers (columns).** The three leftmost columns plot the mean (+/-SEM) number of accumulated salient events in each layer of each perceptual hierarchy as a function of city (blue lines) or office (yellow lines) scene. Only salient events in visual cortex distinguish between office and city scenes, and this holds for all three layers. The three rightmost columns (green lines) plot the difference lines, with shaded bounds depicting 95% CIs. These show that only accumulated salient events in visual cortex distinguish between scenes, because only these lines are above the zero line. This means that accumulated salient events, even prior to regression into standard units (seconds), distinguish between scene type in visual cortex, but not in auditory cortex or in somatosensory cortex.

mapped accumulated events to durations in units of seconds. This means we could use the regression model to further generate predictions of subjective duration biases from salient events in BOLD. It is important to note that the reproduction of presented duration is trivial because, all else being equal, longer intervals will have more salient events in the sensory cortex BOLD dynamics. Indeed, longer intervals will, on average, have a greater number *any* event, task-relevant or otherwise–heart beats, eye movements, planes taking off, etc—and so the cumulative sum of events will trivially correlate with physical duration. Another way to think about this is that taking any summary–the number of salient events, the length of a trajectory through a state space, etc—between a start- and end-point is a temporal metric, and so will correlate with clock time.

Because accumulated salient events will trivially correlate with clock time, testing our proposal necessitates comparing the model of interest (visual cortex) with control models in other modalities (auditory and somatosensory cortex). Contrasting against control models in other modalities allows us to demonstrate that it is not simply the accumulation of any cortical changes over time that predicts duration, but rather accumulation of specific changes in

cortical activity *related to the presented content* that can predict human subjective duration judgements. To reinforce this point, it is because all of our models–visual cortex and the control models based on auditory or somatosensory cortex—do in fact provide reasonable estimates of clock time that our key analyses focus on reproducing the subjective *biases* present in the reports of human participants, since it is these biases that separate clock duration from subjective duration.

Our primary pre-registered hypothesis was that only the visual cortex model would be able to reproduce participants' duration *biases*. Supporting this, only the model trained on visual salient events significantly reproduced the duration biases (calculated from the pooled, human "super-subject" data) trial-by-trial, $\beta_{2328} = 1.51$, $p = 0.015$; the models trained on salient events in auditory cortex, $\beta_{2328} = 0.87$, $p = 0.141$, and somatosensory cortex, $\beta_{2328} = 0.30$, $p = 0.339$, did not (S3 Fig). Using the visual cortex regression beta as our prior [19], evidence for these control model results was insensitive (auditory: $BF_{H(0,1.51)} = 1.22$, RR = $[-\infty, 7.56]$, somatosensory: $BF_{H(0,1.51)} = 0.60$, RR = $[-\infty, 3.14]$).

Not only was the visual cortex regression coefficient a significant *predictor* of behavioral biases, the visual cortex regression model was also a *better fit* to the trial-by-trial behavioral biases than the auditory or somatosensory cortex models (S4 Fig). These results mean that biases in subjective estimates of time can be predicted from neural activity associated with modality-specific perceptual processing. The processing is modality-specific because the video stimuli were silent, with no auditory or tactile stimulation.

While the visual model could reproduce participants' trial-by-trial biases, it did not reproduce the effect of video type (overestimation of duration for city scenes) despite a numerical trend in the predicted direction, $M \pm SD_{diff} = 0.19 \pm 13.96$, 95%CI = $[-0.94, 1.33]$, $t_{2329} = 0.33$, $p = 0.739$, d = 0.01 (S2F Fig). The control models did not reproduce the effect of video type either (auditory: $M \pm SD_{diff} = -0.33 \pm 12.29$, 95%CI = $[-1.32, 0.67]$, $t_{2329} = -0.64$, $p = 0.522$, d = -0.03, somatosensory: $M \pm SD_{diff} = -0.16 \pm 13.09$, 95%CI = $[-1.23, 0.90]$, $t_{2329} = -0.30$, $p = 0.762$, d = -0.01, see S2G–S2H Fig). Setting a uniform prior between 0 and our behavioral effect of scene type (5.23%), Bayes factor analysis found evidence for the null in all three t-tests (visual: $BF_{U(0,5.23)} = 0.18$, RR = $[2.83, \infty]$), auditory: $BF_{U(0,5.23)} = 0.22$, RR = $[3.30, \infty]$), somatosensory: $BF_{U(0,5.23)} = 0.17$, RR = $[2.55, \infty]$)). Note that neither these *t*-tests nor the priors were pre-registered.

## Using signed difference, estimation bias and effects of scene type can be reconstructed from visual cortex BOLD

Next, we analyzed the biases predicted from the exploratory model, in which salient events were determined from signed differences in voxel activity. Again, presented video duration could be (trivially) predicted from salient events in all three exploratory models to a similar degree (10-fold cross validation, $\bar{r}_{visual} = 0.95$, $\bar{r}_{auditory} = 0.97$, $\bar{r}_{somatosensory} = 0.96$, Fig 6A–6C). However, using the revised (exploratory) definition of a salient event, linear mixed models revealed the visual model biases *did* strongly discriminate between office and city scenes, $M \pm SD_{diff} = 4.22 \pm 3.37$, 95% CI = $[3.14, 5.30]$, $\chi^2(1) = 85.06$, $p < .001$ (Fig 6D).

Visual model biases also remained correlated with participants' trial-by-trial biases, $\beta = 0.02 \pm 0.008$, $\chi^2(1) = 5.62$, $p = 0.018$. This association is visualized in Fig 7A by plotting mean model bias as a function of 25 quantiles of human normalized bias. The association held under a wide range of reasonable attention threshold parameters (Fig 7B), meaning that model performance in reproducing participant duration reports was robust to how salient events were categorized. Again, the visual model out-performed control models in predicting normalized bias (S5 Fig). While the model trained on accumulated visual cortex salient events reproduced patterns in

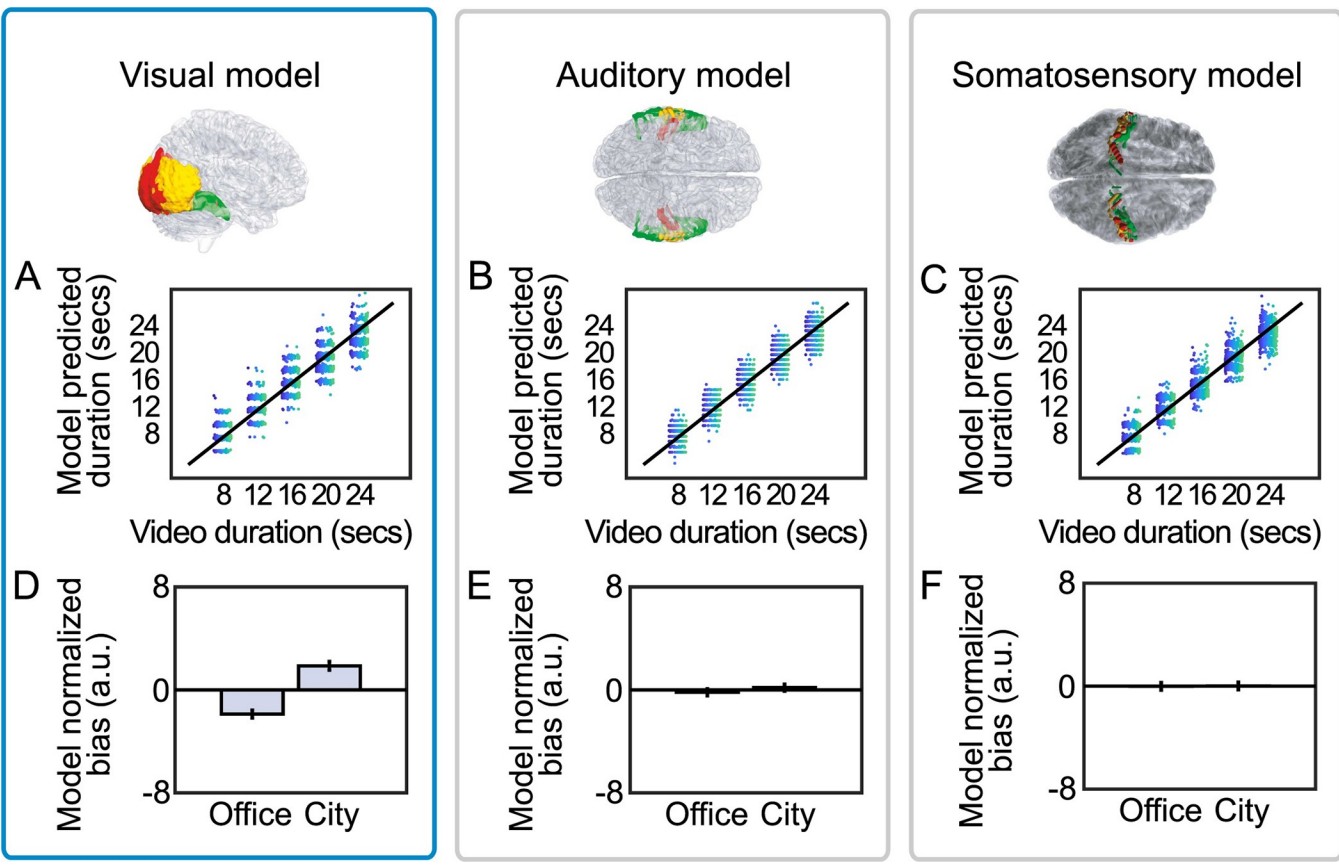

**Fig 6. Computational neuroimaging analysis. (A-C)** Trial-by-trial association between presented video duration and model-predicted duration reports obtained from the visual, auditory, and somatosensory models. Different dot colors represent different participants, and each dot is data from one trial. **(D-F)** Mean model-estimated normalized bias as a function of video type for the visual, auditory, and somatosensory models. Error bars represent +/- within-subject SEM.

human biases, biases from exploratory models trained on auditory and somatosensory salient events did not: they neither discriminated video type (auditory: $M\pm SD_{diff} = 0.35 \pm 2.65$, 95% CI = [-0.50, 1.19], $\chi^2(1) = 0.43$, $p = 0.514$, $BF_{H(0,0.02)} = 0.16$, RR = $[0.02, \infty]$, somatosensory: $M\pm SD_{diff} = -0.32 \pm 2.56$, 95% CI = [-1.13, 0.50], $\chi^2(1) = 0.46$, $p = 0.499$, $BF_{H(0,0.02)} = 0.06$, RR = $[0.01, \infty]$ see Fig 6E and 6F), nor predicted trial-wise human normalized bias (auditory: $\beta = -0.003 \pm 0.006$, $\chi^2(1) = 0.20$, $p = 0.652$, $BF_{H(0,0.04)} = 0.24$, RR = $[0.01, \infty]$, somatosensory: $\beta = 0.002 \pm 0.007$, $\chi^2(1) = 0.11$, $p = 0.740$, $BF_{H(0,0.04)} = 0.46$, RR = $[-\infty, 0.03]$ respectively, Fig 7C and 7E). Note that priors for the Bayes factors were not pre-registered and were set as the fixed-effect coefficients from the corresponding visual cortex LMMs.

The correlational results were also robust across a range of threshold parameters for the visual model (Fig 7B). For the auditory model (Fig 7D), positive correlations between human and model-predicted biases were found only at implausible parameter values (where the threshold's upper bound was the mean). For the somatosensory model (Fig 7F), positive correlations were present in a small, localized region of the space indicating that those correlations were not robust to changes in threshold parameters, and likely spurious or artefactual (e.g., driven by head motion or eye movements).

In none of the auditory or somatosensory layers were there more salient events when watching city than office videos (Fig 5, middle and bottom rows). Further, our ability to predict subjective biases in duration does not trivially follow from differences in the videos

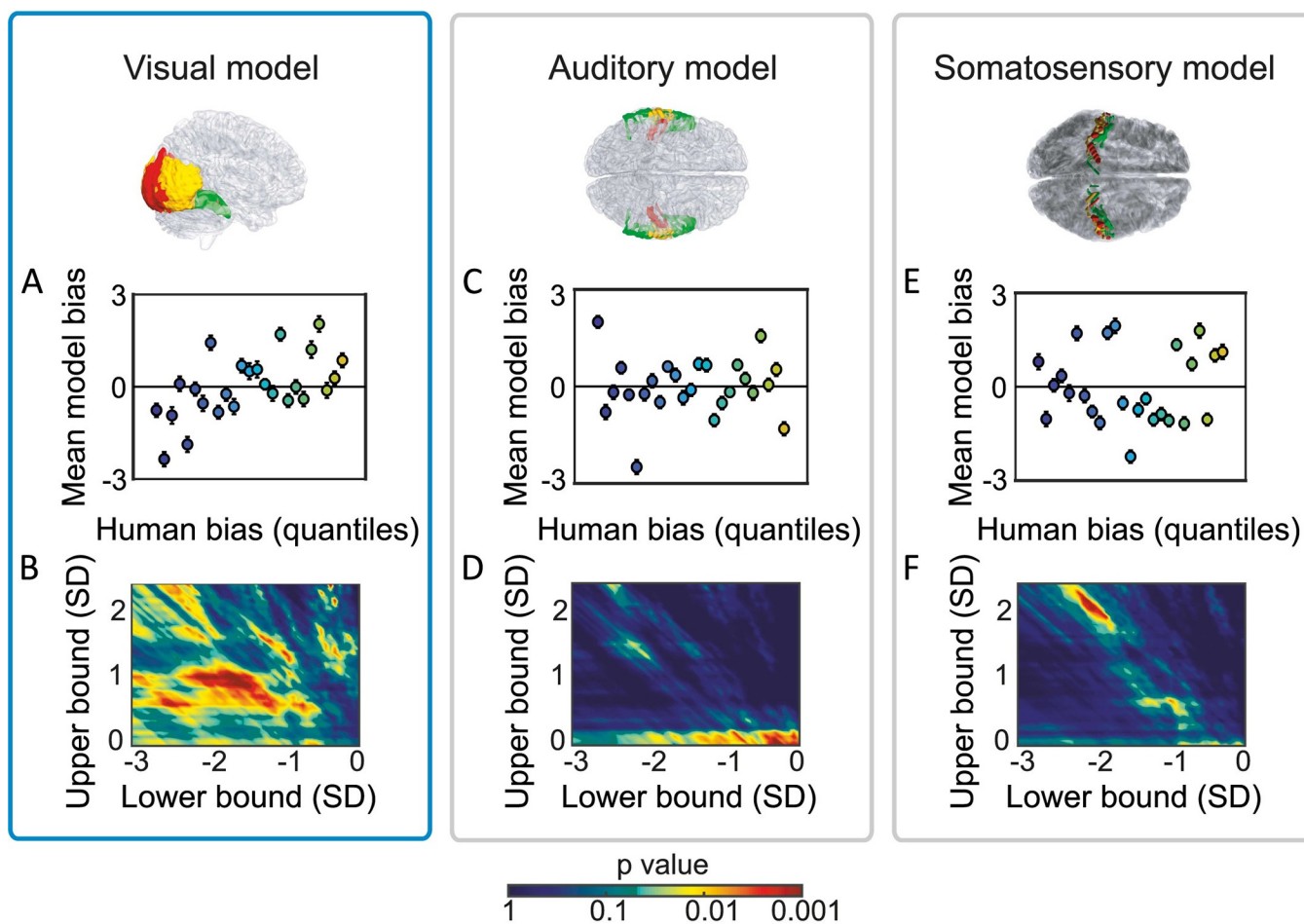

**Fig 7. Predicting trial-by-trial subjective time from human BOLD.** (**A**) Mean normalized bias for the model trained on visual cortex activity, as a function of 25 quantiles of human bias. Colors represent x-axis values. Results show a positive association between human biases and model-predicted biases. (**B**) Heat map depicting p-values for the association between human bias and (visual cortex) model bias, as a function of minimum (x-axis) and maximum (y-axis) criterion values. Dark colors represent regions where the association was non-significant at $\alpha_{0.05}$ or negative. Consistent results are found under a wide range of reasonable parameter values. (**C,E**). As for panel A, but for auditory and somatosensory cortex respectively. There is no association between human bias and model biases. (**D,F**). As for panel B, but for auditory and somatosensory cortex respectively. No significant positive correlation is found under alternative reasonable parameter values.

themselves (e.g. more changes in city than office videos), because frame-by-frame changes in the videos are dissociable from both human and model-predicted biases (see S1 Text and S6 Fig). Taken together, these results underline the specificity of visual cortex activity in predicting subjective time for silent videos.

## Discussion

We have shown that the basis for subjective estimates of duration can be constructed on a trial-by-trial basis from salient events in sensory cortex activity, where salient events are defined as relatively large changes in the neural responses to sensory stimulation across a perceptual hierarchy. Importantly, participants are not necessarily conscious of the events because they are events in the perceptual processing dynamics rather than in the stimulus. Salient events as we have defined them are not necessarily *psychologically* salient either.

In this study, for which stimuli were silent videos, successful prediction was obtained only for models trained on salient events in visual cortex BOLD, and not for control models based

on somatosensory or auditory cortex BOLD. While we could trivially reconstruct clock time from activity in all three sensory regions (because those regions exhibited dynamic neural activity that trivially reflects video duration), only the information extracted by the stimulus-relevant sensory model—the visual model—was related to subjective duration estimates (as reflected in relative over- or under-estimates of duration). Our results were robust under a wide range of model parameter values (Fig 7B), and, in combination with previous findings [7,9], support the idea that human time perception is based in the neural processes associated with processing the sensory content from which time is being judged.

The reconstruction of clock time in all brain-based models was strong (correlations between model estimates and veridical video duration r > .92), and better than our human participants' performance (correlation between participant estimates and veridical video duration r ~ = .8). There are several reasons for this. First, as previously stated, the model predictions will trivially correlate with clock time because more salient events will be accumulated as epoch length increases. Second, we specifically trained these models to transform the number of salient events into clock time (by mapping them to the video durations), and accordingly they do so very well. With feedback, humans can reproduce clock time well also–indeed, even without training participants in this study exhibited a correlation of ~.8 with clock time. Outside of the lab, in more natural cases duration estimation may often be worse: the task is harder, mappings from experience to standard units of report are contextual, feedback may be sparse (and not necessarily even incorporated), and is received over long time scales. In short, by virtue of simultaneous access to all information about the relationships between time sensation (salient events) and clock time, our models (in this case) have an unfair advantage versus humans, who could be doing similar computations (regression), but typically with less complete information.

Our visual cortex ROIs were chosen in such a way that our hierarchy was a highly simplified, "toy model" of human ventral vision. Of course, we by no means assert that other regions in visual cortex are unimportant for duration perception of visual stimuli [20]. Correspondences between stimulus features and neural responses have been shown in regions we have not included, and work that attempts to map the neural responses to temporal properties of visual stimuli has revealed a broad range of temporally responsive regions across the perceptual processing hierarchy, including dorsal visual stream and frontoparietal areas (see e.g. [21]). The present study suggests that for the specific stimulus content we utilized (silent videos) the ROIs we selected were sufficient to predict subjective time, but of course future work could test whether model performance would substantially improve by expanding the set of regions included.

We compared two different metrics for determining change in processing dynamics–Euclidean distance and signed difference, and used dynamic thresholds to classify changes as salient events, with these thresholds allowed to vary across layers of the perceptual hierarchy. We could predict human biases in duration judgements when using both metrics, but could only predict the video class (office scene vs. city scene) from the signed difference. It is notable that even these crude measures are sufficient for mapping processing dynamics within perceptual hierarchies to subjective time. Future work could attempt to optimise the metrics and thresholds according to differences in the known response properties of different visual areas [22,23].

## A novel approach to modelling subjective time

Our model is the first which is able to predict trial-by-trial subjective reports of duration based on neural (BOLD) activity during naturalistic stimulation, and in so doing, advances our

understanding of the neural basis of time perception. Our approach is conceptually related to a study by Ahrens and Sahani [24], who proposed that subjective time perception is constructed from estimates of input dynamics (akin to sensory input) and knowledge of the temporal structure of the input (second order input statistics), and presented an inference model that could account for several behavioural results. An important difference between their work and ours is that statistical "knowledge" in our model relates to knowledge of the perceptual processing network state. By contrast, knowledge in Ahrens and Sahani's model relates to prior temporal structure. This means that while Ahrens and Sahani propose a model dependent on processes dedicated to tracking temporal properties, we do not. Our results demonstrate that such knowledge is not strictly necessary for generating human-like duration estimates for natural stimuli.

### Linking sensory content and subjective duration

In our approach, the neural processes that are engaged in the processing of sensory content (putatively ventral stream vision in our human participants and for our stimuli) are the same as those used to build the basis for estimates of time. In this way we provide an intuitive link between sensory content and subjective duration. Our conclusion is in support of the idea that time perception depends on distributed mechanisms [25], but that in each case subjective time is naturally linked to sensory content by virtue of being determined by those content-related processes.

It is common for studies in the time perception field to use highly constrained stimuli such as luminance contrast discs, Gabors, or random dot fields. The high level of control over stimulation permitted by such stimuli allows these studies to identify one-to-one mappings between a specific stimulus feature (e.g. contrast intensity, temporal frequency, or even stimulus size) and time perception. For example, a presented stimulus is reported as longer in duration in a condition where it is larger in size than in a condition where it is smaller [26]–a simple mapping between the feature of stimulus size and the corresponding duration judgements. A large literature exists showing how recent visual exposure [27,28] or adaptation to temporal properties of a stimulus can influence subjective time judgements [29–31]; that eye movements nearby brief stimuli can change the content of vision, and consequently perception of time [32–34]; how eye movements *during* stimulation change time estimates [35]; and how the precise temporal properties of a visual stimulus–temporal frequency or speed–relate to subjective duration reports [2,3,36]. Previous work has also shown how different parts of the processing hierarchy respond differently to temporal properties of stimuli, such as whether stimuli were composed of transient or persistent stimulation [23]. Further, recent work using direct mapping methods with simple, brief stimuli (intervals up to ~ 2 seconds) has made substantial progress in describing the complex hierarchy of neural responses to temporal properties of visual stimulation [21,37].

The existing time perception literature, as described above, has been successful in characterising the relationship between visual response properties and subjective time. Our study extends this work by taking two key differences in approach. First, we did not directly look for a correspondence between temporal properties of stimulation and neural responses reflecting those temporal properties (i.e. we didn't look for neural regions that represent the temporal properties of the stimulation–'chronotopy'). Instead, we attempted to build a model that extracts relevant temporal properties from the neural activity associated with processing of perceptual content and produces estimates of duration based on that information–an indirect rather than direct mapping between the stimulus and neural response to stimulus on the one hand, and subjective duration on the other. The second key difference in our approach comes

from our use of naturalistic scenes versus the simple, highly-controlled stimuli frequently used. Natural scenes contain a wide variety and combination of simple and complex features, some of which will be coded for in overlapping neural populations. These features vary in their combinations moment to moment, and so to build a successful model of time perception for *natural* stimulation it is necessary to consider their joint contributions. By using a measure based on extracted *salient events* across the perceptual processing hierarchy, our approach accomplishes this task without needing to fully characterise those joint contributions. We note that to do so would be a very difficult task indeed. While a complete understanding of how these features are identified, tracked and combined in cortex remains an important challenge for visual neuroscience, and progress has been made (as we review above), our approach is able to jointly accommodate the many stimulus features that vary during natural stimulation in a simple fashion, without needing to identify and track each pertinent feature independently, and then combine them in such a way that respects their interdependence (e.g. contrast drifts during configural changes).

There are interesting opportunities for future work at a greater level of specificity, which could attempt to fully specify the correspondences between all simple features and subjective duration, as well as their combinations. If successful, this detailed approach would allow very strong predictions about the precise, moment-to-moment contribution of some specific stimulus feature (and neural response to that specific feature) to subjective time perception. However, as our model demonstrates, having this complete understanding is not necessary for successfully modelling human distortions in subjective time.

### Attention, emotion and time

In the time perception literature there are often appeals to the influence of factors like attention [38] or emotion [39,40]–sometimes used as the basis for rejecting the proposals of earlier cognitive psychologists such as Ornstein [5]. Our model provides a general basis on which to test any claims about any such influences by specifying a baseline hypothesis in each case–namely that the dynamics of the relevant sensory cortex (e.g. visual, auditory, interoceptive, etc) are sufficient to construct subjective duration estimates for that context. For example, regarding the potential influence of emotion on time perception [39,40], the degree to which stimulus-driven differences in network activation underlie any differences in duration estimates remains to be established. However, even if influences of emotion on time perception arise from internally generated sources (rather than processing of external stimulation), this influence may still be reflected in differences in measurable activity of perceptual processing networks (via e.g. BOLD) and therefore our model would reproduce differences in human duration reports. These possibilities are testable hypotheses, made available by our modelling approach.

### Time perception for non-visual or multisensory cases

While we only tested whether subjective duration for visual stimuli could be constructed from salient events in visual cortex, we expect that salient events from auditory cortex would predict subjective time in auditory-only contexts, and similarly for other modalities [41]. Outside the laboratory we judge time in multisensory contexts and can estimate duration when our eyes are closed or even if clinically deaf. These observations do not expose a weakness of our approach, but generate specific and testable claims that require additional study to fully evaluate: does our model approach work equally well in other (combinations of) modalities? We see no reason why our proposal should fail for other modalities, and the success of similar network architectures for interpreting sensory processing in the auditory domain [42] supports this

position. Furthermore, because we define 'salient events' as events in the dynamics of perceptual processing rather than in the external world, non-external stimulation like visual imagery could also contribute to experience of time. The model we present here is based on unimodal inputs, but for multisensory inputs we would predict that duration biases would be combined through some integration of accumulated salient events in modality-specific sensory cortices (e.g. for videos with sound, an integration of accumulated salient events in visual and auditory cortices). This may broadly follow what is understood about combination in other multisensory cases and may be consistent with Bayesian rules of combination [43,44].

It could be suggested that our approach only reproduces stimulus-driven biases rather than providing a general basis for time estimation because without stimulus input our model would have no "activity". This critique would be valid for the artificial network-based model in section *Estimates generated by an artificial network model are biased by scene type* but cannot be applied to the BOLD-based model because visual cortex activity remains present even when there is no sensory input (see also above point about imagery).

## Biological plausibility

Our data do not speak to the question of *how* perceptual processing is achieved by the brain, and our results do not depend on the answer, beyond some key, uncontroversial assumptions (e.g. hierarchical processing). Whether the classification network used here (AlexNet) is closely matched to biological vision in how it processes information is not relevant here (see [13–15]) because the algorithmic approach to estimating duration from network activity (in either AlexNet or estimated from BOLD in humans) produces outcomes consistent with the patterns seen in human subjective reports of time. Accordingly, we did not directly compare the predictions from the classification network and brain-based models. The crucial assumption is simply the existence of a hierarchical, specialized system for perceptual processing—the common interpretation of primate ventral visual stream [45–47]. Given this assumption, our model is compatible with a range of prominent theories on the specific computational processes that may contribute to how perception is achieved (predictive coding [48], population coding [49], Gibsonian affordances [50] etc.), and the mechanisms underlying how feature extraction/processing occurs (population receptive fields [51], feedback connections [52], surround suppression [53] etc). Our claim is simply that the dynamics of perceptual systems can be used to construct subjective duration, but is theory-neutral as to precisely which processes are most important for perception. This conclusion is best demonstrated by the fact that our model produced estimates consistent with subjective biases in human reports regardless of whether applied to activation patterns of AlexNet or to BOLD patterns recorded from human participants.

## Predictive processing as a potential mechanistic basis for time perception

While we are theory-neutral regarding the specific neural basis of perceptual processing, our results may provide some evidence in favour of one potential mechanistic basis for time perception. We tested two metrics that could be used by the brain to link sensory content and time on a moment-to-moment basis: Euclidean distance (pre-registered) and signed difference (exploratory). Whereas the former assumes that BOLD activity indexes some raw quantity associated with sensory inputs, the latter assumes that BOLD already indexes change in sensory input, for example as perceptual prediction error. In our data, subjective duration was best reconstructed using signed difference: although both metrics generated duration estimates that correlated with human reports, only signed difference differentiated video type. The superiority of signed difference in predicting subjective time is consistent with (but not evidence

for) the view that BOLD already indexes detected environmental changes. This is in line with literature evidencing "surprise" or "prediction error" responses in sensory [18,54,55] and even frontal [56,57] cortices, usually interpreted in the context of predictive processing [58] or predictive coding [42] theories of cortical function. Of course, this superiority of signed difference is not itself evidence for a role for prediction error in time perception, nor are the theories of predictive processing [58] or predictive coding [48] necessary for understanding or interpreting our results.

We also emphasize that the way in which we use "salience" and "surprise" is only tangentially, if at all, related to the psychological phenomena of something being salient or surprising. Here, salience is defined in terms of difference between successive network states (see Eqs 2 and 3). This means our notion of salience is close to a naïve prediction error [6]; naïve because the "prediction" is simply the previous network state rather than part of a prediction-update cycle (see [7]). While previous studies have suggested that predictability [59] or apparent salience [60] can affect subjective time perception [61], descriptions of "salience" and related terms at this cognitive level are not necessarily related to descriptions at the mechanistic level at which our model is articulated. Future work may wish to test whether "prediction error" as defined in a mechanistic sense maps onto psychological salience or surprise, but the question is outside the scope of the present study, and is certainly not restricted to investigations of time perception.

### "Surprise", time perception, and episodic memory

The idea that our model may be based on an index of perceptual "surprise" is intriguing as it provides a natural link to the closely related topic of episodic memory (see [7]). In the episodic memory literature, prediction error, i.e. the difference between current sensory stimulation and expected stimulation, has been proposed as the basis for the construction of event boundaries [7,17,62]–transitions that segment some content (e.g. a cow) from some other content (e.g. a car) in continuous experience [63,64]. By emphasizing the importance of sensory content in time perception, our approach may provide a link between time perception and episodic memory that has been lost by content-free "clock" approaches. By providing a simple algorithm for how the stream of sensory processing is segmented into salient events, our approach may afford some insight into how low-level sensory information is transformed into the temporally sequenced form of memory associated with the activity of so-called "time cells", potentially linking the content of sensory processing with temporal properties of episodic memory within the powerful predictive coding approach [7,48,65].

### Conclusions

In summary, we provide evidence for an algorithmic account of duration perception, in which information sufficient for the basis of subjective time estimation can be obtained simply by tracking the dynamics of the relevant perceptual processing hierarchy. In this view, the processes underlying subjective time have their neural substrates in perceptual and memory systems, not in systems specialized for time itself. We have taken a model-based approach to describe how sensory information arriving in primary sensory areas is transformed into subjective time, and tested this approach against human neuroimaging data. Our model provides a computational basis from which we can unravel how human subjective time is generated, encompassing every step from sensory processing to the detection of salient perceptual events, and potentially further on to the construction and ordering of episodic memory.

## Materials and methods

### Ethics statement

The study was approved by the Brighton and Sussex Medical School Research Governance and Ethics Committee (reference number ERA/MS547/17/1). All participants gave informed, written consent and were reimbursed £15 for their time.

### Participants

Forty healthy, English speaking and right-handed participants were tested (18–43 years old, mean age = 22y 10mo, 26 females). Sample size was determined according to funding availability.

### Procedure

The experiment was conducted in one sixty-minute session. Participants were placed in the scanner and viewed a computer visual display via a head-mounted eyetracker, placed over a 64-channel head coil. Eyetracker calibration lasted approximately five minutes and involved participants tracking a black, shrinking dot across nine locations: in the center, corners and sides of the visual display. Eyetracking data are not used in this manuscript due to technical failure.

Following calibration, we acquired six images reflecting distortions in the magnetic field (three in each of the posterior-to-anterior and anterior-to-posterior directions) and one T1-weighted structural scan.

Finally, functional echoplanar images (EPIs) were acquired while participants performed two to four blocks (time-permitting) of twenty trials, in which participants viewed silent videos of variable length and reported the duration of each video using a visual analogue scale extending from 0 to 40 seconds (see Fig 1A). A key grip was placed in each hand, and participants moved a slider left and right using a key press with the corresponding hand. The initial position of the slider was randomised trial-by-trial. Participants were not trained on the task prior to the experimental session.

### Experimental design and trial sequence

Each experimental block consisted of 20 trials. On each trial a video of duration 8, 12, 16, 20 or 24 seconds was presented. For each participant, videos of the appropriate duration and scene category were constructed by randomly sampling continuous frames from the stimuli built for [6]. These videos depicted either an office scene or a city scene. Two videos for each duration and content condition were presented per block in randomized order. For one participant and one block, only 11/20 trials were completed giving a total of 2331 trials across the entire dataset.

### MRI acquisition and pre-processing (confirmatory)

Functional T2$^*$ sensitive multi-band echoplanar images (EPIs) were acquired on a Siemens PRISMA 3T scanner (2mm slices with 2mm gaps, TR = 800ms, multiband factor = 8, TE = 37ms, Flip angle = 52˚). To minimize signal dropout from parietal, motor and occipital cortices, axial slices were tilted. Full brain T1-weighted structural scans were acquired on the same scanner using the MPRAGE protocol and consisting of 176 1mm thick sagittal slices (TR = 2730ms, TE = 3.57ms, FOV = 224mm x 256mm, Flip angle = 52˚). Finally, we collected reverse-phase spin echo field maps, with three volumes for each of the posterior to anterior and anterior to posterior directions (TR = 8000ms, TE = 66ms, Flip Angle = 90˚). Corrections

for field distortions were applied by building fieldmaps from the two phase-encoded image sets using FSL's TOPUP function. All other image pre-processing was conducted using SPM12 (http://www.fil.ion.ucl.ac.uk/spm/software/spm12).

The first four functional volumes of each run were treated as dummy scans and discarded. A standard image pre-processing pipeline was used: anatomical and functional images were reoriented to the anterior commissure; EPIs were aligned to each other, unwarped using the fieldmaps, and co-registered to the structural scan by minimizing normalized mutual information. Note that in accordance with HCP guidelines for multiband fMRI we did not perform slice-time correction [66]. After co-registration, EPIs were spatially normalized to MNI space using parameters obtained from the segmentation of T1 images into grey and white matter, then smoothed with a 4mm FWHM Gaussian smoothing kernel. Smoothed data were used for the GLM on BOLD only; unsmoothed data were used for the brain-based modelling.

## Statistical analyses

All fMRI pre-processing, participant exclusion criteria, behavioral, imaging and computational analyses were comprehensively pre-registered while data collection was ongoing (osf.io/ce9tp/) but before it was completed. This analysis plan was determined based on pilot data from four participants, and was written blind to the data included in this manuscript. Analyses that deviate from the pre-registered analysis plan are marked as "exploratory". Pre-registered analyses are described as "confirmatory". Data are freely available to download at https://osf.io/2zqfu.

## fMRI statistical analysis (confirmatory)

At the participant level, BOLD responses obtained from the smoothed images were time-locked to video onset. BOLD responses were modelled by convolving the canonical hemodynamic response function with a boxcar function (representing video presentation) with width equal to video duration. Videos of office and city scenes were modelled using one dummy-coded regressor each. Each was parametrically modulated by normalized bias.

Data from each run was entered separately. No band-pass filter was applied. Instead, low-frequency drifts were regressed out by entering white matter drift (averaged over the brain) as a nuisance regressor [57,67]. Nuisance regressors representing the experimental run and six head motion parameters were also included in the first level models. Because of the fast TR, models were estimated using the 'FAST' method implemented in SPM.

Comparisons of interest were tested by running four one-sample t-tests against zero at the participant level for each variable of interest (video scenes, office scenes, and their normalized bias parametric modulator). Next, group-level F tests were run on those one-sample contrast images to test for effects of video type and the interaction between video type and normalized bias slope. A one-sample t-test against zero at the group level tested the slope of the normalized bias-BOLD relationship. All group-level contrasts were run with peak thresholds of $p < .001$ (uncorrected) and corrected for multiple comparisons at the cluster level using the FWE method. Clusters were labelled using WFU PickAtlas software [68,69].

## Model-based fMRI (confirmatory)

Our key prediction was that subjective duration estimates (for these silent videos) arise from the accumulation of salient (perceptual) events detected by the visual system, particularly within higher-level regions related to object processing. We tested this by defining a (pre-registered) three-layer hierarchy of regions to represent core features of the visual system:

Layer 1 was defined as bilateral V1, V2v and V3v, Layer 2 was defined as bilateral hV4, LO1 and LO2, and Layer 3 as bilateral VO1, VO2, PHC1 and PHC2 (clusters are depicted in Fig 3).

For each layer, masks were constructed by combining voxels from each area, using the atlas presented in [70].

To determine events detected by the visual system over the course of each video, we extracted raw voxel activity for each TR in each layer from unsmoothed, normalized EPIs. Then, for each voxel v, change was defined as the Euclidean distance between BOLD activation $x_v$ at volume TR and TR-1. The amount of change detected by the layer at any time point, denoted $\Delta_{TR}$, was then given by summing the Euclidean distances over all voxels such that:

$$\Delta_{TR} = \sum_v |X_{TR,v} - X_{TR-1,v}| \tag{2}$$

This process furnishes one value per layer for each TR of each trial for each participant. The next step was to categorize each value as a "salient" event or not and convert it to an estimate of duration using an event detection, accumulation and regression model, as presented in Roseboom et al. [6]. Before converting accumulated salient changes to units of seconds, we first pooled participants' data by z-scoring the summed events $\Delta_{TR}$ within each participant and layer. Pooling was performed to increase statistical power of subsequent regression analyses. Then, for each trial, TR-by-TR categorization of $\Delta_{TR}$ was achieved by comparing against a criterion with exponential decay, corrupted by Gaussian noise $\varepsilon$:

$$\vartheta_{TR} = ae^{-TR} + \varepsilon, \ \varepsilon \sim \mathcal{N}(0, 0.05) \tag{4}$$

We chose the same criterion function used in [6]: while they found that constant (not decaying) thresholds can also produce human-like biases in duration estimates, the decaying threshold reflects the intuitive notion that classification of what is salient should adjust with fluctuations in the environment.

Only the parameter a took different values in each layer (see S2 Table): it took larger values at higher layers. In this way, the thresholds could accommodate different types of scene, e.g. scenes with more high-level configural changes, or scenes with more low-level changes. The criterion decayed with each TR until either an event was classified as salient or until the video finished, after each of which the criterion reset to its starting (i.e. maximal) point. Importantly, because the summed Euclidean distances $\Delta_{TR}$ were z-scored, the criterion has meaningful units corresponding to SDs above or below the mean. The parameter *a* corresponds to the largest z-score above which a change was classified as salient, that is, the criterion's most conservative point. To account for potential head-motion artefacts, criterion updating ignored volumes where $\Delta_{TR}$ was greater than 2.5 (i.e. more than 2.5 SDs from the mean).

The final modelling step was to convert the BOLD-determined accumulation of salient events into raw duration judgements (in seconds). This was achieved via Epsilon-support vector regression (SVR), implemented on python 3.0 using *sklearn* [71], to regress accumulated events in each of the three layers onto the duration of the presented video.

To evaluate whether the model could reproduce subjective reports of time from participants' BOLD activation, we converted the trial-by-trial model predictions (raw duration judgements in seconds) to normalized bias. These were then compared to a human "super-subject": participants' duration judgements were z-scored within participants, then all participant data were pooled and converted to normalized bias. We created a super-subject to mirror the data pooling performed before training our SVR.

Trial-by-trial normalized bias values were compared across model and human using linear regression, fitting the model:

$$behaviour_t = \beta_0 + \beta_1 model_t \tag{5}$$

To test our *a priori* hypothesis that the model trained on visual cortex salient events positively correlates with subjective time, a (one-tailed) p-value for $\beta_1$ was calculated via bootstrapping, shuffling the behavioural data and refitting the regression line 10,000 times.

## Control models (confirmatory)

To distinguish our proposal from the more trivial suggestion that the neural dynamics of any cortical hierarchy (or any neural ensemble) can be used to approximate elapsed clock time, simply because they are dynamic, we created two control models. While these models should all approximately reproduce clock time, the reproduced estimates should not be predictive of the specifically subjective aspects human participants' duration estimates (i.e., their biases). Analyses for these control hierarchies followed the steps above for the primary model, though based on different sensory regions.

The first control hierarchy was auditory cortex, which has previously been implicated in time perception but whose involvement in duration judgements should not be driven by visual stimuli, as in our study. Layers 1 and 2 were defined as Brodmann Area (BA) 41 and 42 respectively, both of which are located in primary auditory cortex. Layer 3 was posterior BA22 (superior temporal gyrus/Wernicke's Area).

The second control hierarchy was somatosensory cortex, which on our model should not be involved in duration judgements based on visual stimuli. Layer 1 was set as posterior and anterior BA 3, and layers 2 and 3 were set as BA 1 and BA 2 respectively. These Brodmann areas correspond to the primary somatosensory cortex.

Masks for these two control analyses were constructed using WFU PickAtlas atlases [68,69]. As for our empirical analyses using visual cortex, for each of the two controls we estimated the relationship between the trial-by-trial normalized bias based on the model's predictions and based on z-scored participant data by fitting a linear regression line.

To test whether the visual cortex model out-performed the somatosensory and auditory cortex models we compared their log-likelihoods, obtained from the Matlab function *fitlm* (see S4 Fig). This evaluation of model performance was not pre-registered.

## Exploratory modelling

We also ran an exploratory (i.e. not pre-registered) set of models. This was identical to the pre-registered analysis plan, apart from the following differences:

First, we transformed voxel-wise BOLD activation X to signed (i.e. raw) rather than unsigned changes:

$$\Delta'_{TR} = \sum_v (X_{TR,v} - X_{TR-1,v}) \tag{3}$$

Using SVR as before, for each hierarchy we obtained model-predicted duration estimates in seconds. To avoid pooling participants' reports together, human judgements were not standardized. Instead, for each of our 40 participants we computed human and model normalized biases from the human reports and model predictions associated with the set of videos associated with each participant. In other words, normalized bias was computed 'within-participant'.

To test the association between video-by-video human and model bias while accounting within-participant variability we used a linear mixed model approach. Using R with the *lmer* and *car* packages, we fit the following random-intercept model:

$$bias_{human} \sim 1 + bias_{model} + (1|participant) \tag{6}$$

To determine whether model ($bias_{model}$) and human ($bias_{human}$) biases correlate, we used a chi-squared test (from the *car* function *Anova*) to compare Eq 5 to a reduced model without the fixed effect:

$$bias_{human} \sim 1 + (1|participant) \tag{7}$$

To test the effect of video type (or scene) on model normalized bias, we fit the model:

$$bias_{human} \sim 1 + scene + (1|participant) \tag{8}$$

Again, we used a chi-squared test to compare Eq 8 to the reduced model that did not include *scene* (Eq 7)

To test whether the model trained on visual cortex events outperformed the somatosensory and auditory models, we compared the difference in AIC between the main (Eq 6 and Eq 8) and control (Eq 7) models for each hierarchy (see S5 Fig).

## Robustness analysis (exploratory)

To examine the robustness of our exploratory analysis to criterion parameters we reran the above analysis pipeline under varying values of $\vartheta_{min}$ and $\vartheta_{max}$. For layer 1 (where there should be most salient changes), $\vartheta_{min}$ took 50 linearly-spaced values between 3 SD and 0 SD below the mean. $\vartheta_{max}$ independently took 50 linearly-spaced values between 0 SD and 2.5 SD above the mean. We chose 2.5 SD because this was the highest value z-scored BOLD could take before being discarded as a head motion artefact. For each pair of $\vartheta_{min}$ and $\vartheta_{max}$ values for layer 1, the lower/upper bounds for layer 2 were $\vartheta_{min} + 0.5$ and $\vartheta_{max} = 0.5$ respectively. For layer 3, they were $\vartheta_{min} + 1$ and $\vartheta_{max} + 1$ respectively.

With these criteria, we obtained 250 datasets for each ROI. For each ROI and dataset, we tested the association between model-predicted bias and human bias by fitting the regression model:

$$bias_{human} \sim \beta_0 + \beta_1 \times bias_{model} \tag{9}$$

Heat maps depicted in Fig 7 correspond to one-tailed p-values for $\beta_1$. This robustness analysis was not pre-registered.

## Artificial classification network-based modelling

Frames from each video presented during the experiment were fed into the model presented in Roseboom et al [6]. Instead of accumulating events based on changes in BOLD amplitude, salient events in the video frames themselves were detected by analyzing activity in an artificial image classification network (AlexNet)[16]. We used nine network layers (input, conv1, conv2, conv3, conv4, conv5, fc6, fc7, and output, where fc corresponds to a fully connected layer and conv to the combination of a convolutional and a max pooling layer). Node-wise Euclidean distances for each node were computed, then summed over all nodes in the layer giving one value per video frame and layer. Each value was classified as a salient event or not using the same exponentially decaying criterion as before (see S3 Table for criterion values). Finally, accumulated salient events were mapped onto units of seconds using multiple linear regression.

## Supporting information

**S1 Fig. Results from confirmatory GLM on BOLD (significant clusters only). A** Higher BOLD for city than office scenes: R lingual gyrus; bilateral midcingulate area; R insula; bilateral

SFG. **B** Higher BOLD for office than city scenes: R precuneus; bilateral precentral gyrus; L MFG; bilateral cerebellum; L paracentral lobule; R SFG. **C** Positive correlation with normalized estimation bias: bilateral precentral gyrus; L SMA; R superior occipital gyrus. **D** Negative correlation with normalized estimation bias: L angular frontal gyrus; L MFG; L posterior cingulate. See also S2 Table.
(TIF)

**S2 Fig. Brain-based modelling on the pre-registered pipeline. (A)** Strong positive association between presented video durations and the z-scored reports we used to build the super-subject. **(B)** Normalized estimation bias computed on pooled ('super-subject') behavioral data, as a function of video scene. **(C-E)** Association between presented video duration and model-predicted durations separately for visual, auditory and somatosensory Euclidean Distance models respectively. **(F-H)** Mean normalized bias of the visual, auditory and somatosensory models respectively, for office versus city scenes. Dot colors in the scatterplots represent different participants. Error bars in the bar charts represent SEM.
(TIF)

**S3 Fig.** *Normalized bias predicted by models trained on salient events (Euclidean distance) in (A) visual, (B) auditory and (C) somatosensory hierarchies.* On the x-axis is the 25 bins representing 25 quantiles of human super-subject bias, and on the y-axis is mean model bias for the trials that fell within in the respective bins. Error bars represent +/- SEM.
(TIF)

**S4 Fig. Model fits for the regression of predicted durations onto presented durations (Euclidean Distance models), expressed as log-likelihood ratios. For each of the visual, auditory and somatosensory models, we regressed the model-predicted biases onto the human super-subject's biases.** To compare the performance of the three regressions we compared their log-likelihoods to the null (intercept) model (higher values indicate better model fits). The visual cortex regression outperforms the other two, as indicated by its higher log-likelihood ratio.
(TIF)

**S5 Fig. Model fits for the linear mixed models (Signed Difference analyses). Left.** To test whether the visual, auditory or somatosensory models generated predicted durations that discriminated video type, we ran linear mixed models (LMMs) predicting model biases from the fixed effect video scene (city vs office). These were compared to control LMMs that did not have this fixed effect, using the log-likelihood ratio (LLR). The visual cortex LMM outperformed the auditory and somatosensory cortex LMMs as indicated by the greater LLR. **Right.** For each of the visual, auditory and somatosensory models, we constructed an LMM with human bias as the outcome and the model-predicted biases as a fixed effect. These LMMs tested the video-by-video correlations between predicted and human bias. These LMMs were compared to control models that did not have the model-predicted bias as a fixed effect, using LLR. The visual cortex LMM outperformed the auditory and somatosensory cortex LMMs, as indicated by the greater LLR.
(TIF)

**S6 Fig. Dissociation between human/fMRI-model duration estimates and stimulus properties.** We identified pairs of trials from the same participant, but from different video categories for which **(A)** human reports were very similar (the log ratio did not exceed 0.025) or **(B)** the reports predicted by the visual cortex model were very similar (the log ratio did not exceed 0.025). In both panels A and B, a dot represents a pair of trials. Dot colour represents the

participant. For each pair, differences in report (human in panel A or model-predicted in panel B) are plotted against differences in the physical video differences, here quantified as the frame-to-frame Euclidean distance averaged over pixels. The difference in report/Euclidean distance between the two trials in a pair is expressed as log(city/office). These figures show that there were many trials pairs in our data where, despite being very different in terms of the pixel differences (up to 100s of times), human duration estimations (A) and visual cortex-based model predictions (B) were almost identical.
(TIF)

**S1 Text. Supplementary results.** Pixel-wise changes in stimulation are dissociable from both human and model-predicted report.
(DOC)

**S1 Table. Definition of hierarchies for each sensory cortex model.**
(PDF)

**S2 Table. Criterion parameters for each hierarchical layer of the sensory cortex models.**
(PDF)

**S3 Table. Criterion parameters for the artificial network model.**
(PDF)

**S4 Table. Significant clusters revealed by confirmatory GLM on BOLD.**
(PDF)

## Acknowledgments

Thank you to Charlotte Rae, Petar Raykov, Samira Bouyagoub, Chris Bird, Francesca Simonelli, and Mara Cercignani for their assistance with this project. Thanks also to Virginie van Wassenhove and Martin Wiener for comments on an earlier version of the manuscript.

## Author Contributions

**Conceptualization:** Maxine T. Sherman, Zafeirios Fountas, Warrick Roseboom.

**Data curation:** Maxine T. Sherman.

**Formal analysis:** Maxine T. Sherman, Warrick Roseboom.

**Funding acquisition:** Anil K. Seth.

**Investigation:** Maxine T. Sherman, Zafeirios Fountas, Warrick Roseboom.

**Methodology:** Maxine T. Sherman, Zafeirios Fountas, Warrick Roseboom.

**Project administration:** Maxine T. Sherman, Anil K. Seth, Warrick Roseboom.

**Resources:** Maxine T. Sherman, Zafeirios Fountas, Warrick Roseboom.

**Software:** Maxine T. Sherman, Zafeirios Fountas, Warrick Roseboom.

**Supervision:** Warrick Roseboom.

**Validation:** Maxine T. Sherman, Warrick Roseboom.

**Visualization:** Maxine T. Sherman, Warrick Roseboom.

**Writing – original draft:** Maxine T. Sherman, Warrick Roseboom.

**Writing – review & editing:** Maxine T. Sherman, Zafeirios Fountas, Anil K. Seth, Warrick Roseboom.

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
