## [Decision Letter · Decision Letter 0]

17 Jan 2022

Dear Dr Roseboom,

Thank you very much for submitting your manuscript "Trial-by-trial predictions of subjective time from human brain activity" for consideration at PLOS Computational Biology.

As with all papers reviewed by the journal, your manuscript was reviewed by members of the editorial board and by several independent reviewers. In light of the reviews (below this email), we would like to invite the resubmission of a significantly-revised version that takes into account the reviewers' comments.

We cannot make any decision about publication until we have seen the revised manuscript and your response to the reviewers' comments. Your revised manuscript is also likely to be sent to reviewers for further evaluation.

Sincerely,

Ming Bo Cai

Associate Editor

PLOS Computational Biology

Daniele Marinazzo

Deputy Editor

PLOS Computational Biology

Reviewer's Responses to Questions

**Comments to the Authors:**

Reviewer #1: Sherman et al provide a generally strong study of how the perceived duration of natural scene videos is correlated with the activity of the early visual cortex (measured with fMRI) and deep learning neural networks. Most importantly, they find that trial-by-trial variations in perceived duration follow the activity of early visual cortex. This finding supports the view that perception of a sensory stimulus’s duration reflects the activity of the sensory areas processing that stimulus rather than some centralised internal clock mechanisms that is often assumed to underlie time perception. Similar proposals are made by other recent studies that examine the changes in early visual cortex responses with event timing. The novelty of this study focusses on linking neural activity to perceived stimulus duration (rather than veridical stimulus duration) and using natural scenes like we experience in our daily lives (rather than the artificial, mathematically defined but easily controlled stimuli typically used in psychophysics).

Overall, I feel that the experiments provide clear results and careful controls are used. However, the interpretation of these results is complex, because natural scenes are not easily controlled for stimulus properties. While some effort is made to test the possibility that low-level visual features of the scenes underlie the visual cortex responses, these control analyses do not at all follow the known response properties of the visual system. This can be seen as a major shortcoming because it is exactly the responses properties of the visual system that are the focus of this study. But I feel that major changes to these control analyses are beyond the scope of this study, and major further analyses are not needed. Instead the problem is that the discussion of the known response properties of the visual system is lacking and rather dismissive, while in fact there are very good existing models of visual response properties, how visual responses are affected by stimulus properties, and how time perception is related to stimulus properties and the responses of the visual system. These should be discussed properly, constructively, and with reference to current models the visual cortex’s response properties.

Major:

Specifically, Figure 7A convincingly demonstrates the accumulation of neural responses to salient visual events predicts the trial-by-trial bias in perceived movie duration. Figure 8A then shows that there is no relationship between the Euclidean distance between pixels and human perceived duration. The problem with this analysis is that even an undergraduate textbook would make clear that the visual system does not simply respond to changes in pixel colour, and there is a whole literature on what the visual system does respond to. So, while this is a control for the possibility that duration perception depends on low-level stimulus properties, it is not well founded in the literature and is essentially a straw man argument: “it is not the changes detectable in the videos themselves that is key, but the response of perceptual classification networks”. This does nothing to rule out a role of properties of the videos that are understood to underlie visual responses, like contrast, spatial frequency, temporal frequency or the changes in these properties over space and time. I do not think these properties need to be analysed (my third comment will clarify why not), but they do need to be discussed when considering what underlies the responses of the visual system.

I do not think it is enough to say that natural scene movies vary in many complex ways, and so write as if these variations are better ignored when studying natural scene movie perception. I have a lot of trouble with the following statement: “By using a metric based on salient events across the perceptual processing hierarchy, our approach accomplishes this task, allowing us to model natural stimulation in a way where tracking only a single stimulus feature would fail (e.g. tracking only luminance contrast intensity in natural scenes would not predict time perception for scenes where little luminance change occurs; tracking only motion energy would not predict time perception for scenes where there is little movement).” I appreciate the value of a hierarchical network in finding these combinations, but we can’t say that a model based on a single feature would fail unless we try. The authors do not try. This type of statement is found throughout, and comes across as very dismissive of previous studies that use this approach to reveal important things about time perception (see the list towards the end of my review). I encourage a more constructive attempt to reconcile the current findings with previous literature.

At the other extreme, the authors test the predictions of deep convolutional neural network models. Here they find that these models do predict early visual responses and movie duration perception. While such neural network models are often seen as a black box, actually we understand that early network layers filter image inputs to detect contrast at particular orientations and spatial frequencies while middle levels analyse how the outputs of these filters change over the image. These are exactly the stimulus properties that determine how the visual system responds, so the current analyses strongly suggest that such mid-level stimulus features underlie that activation of the visual system and also duration perception. Again, I therefore don’t think it is enough to say that neural network responses are complex, treat them as a black box, and so write as if known visual response properties can be ignored. Again, as a study of the response properties of the visual system, I feel it is very important to discuss what we already know about the response properties of the visual system and try to bring the results of this study into line with that framework, even if this might have to be rather speculative without conducting a lot of further controls that are not necessary for interpreting the current experiments.

There is one aspect of the deep network model I find puzzling. The model used is a feedforward model trained for object recognition, so models the ventral visual stream processing still images. On the other hand, the brain areas that seem to be most involved in representing visual stimulus timing are in the dorsal stream and frontoparietal network (and the early visual system, which is common to both dorsal and ventral streams) (see Hayashi et al 2018, Protopapa et al 2019, Harvey et al 2020, cited below). It’s fine to use AlexNet as an easily available, existing model, but some strange conclusions are drawn as a result of this choice. Notably: “the response of perceptual classification networks … allows us to predict human reports of subjective time”. Indeed the network used is trained for classification, but there is no effort to demonstrate that training for classification is required or that the ventral stream is important in the neural responses seen. Instead, it seems just as likely that it is the early stages of the network are picking up on the relevant movie features and would do so in a generative network that has not been trained for classification at all. Therefore, I feel that conclusions based on the structure (hierarchical, convolutional, feedforward) or training (classification, object processing) of the specific network used are not supported because no other networks were tested. These conclusions are not central to this study, but the role of classification is mentioned repeatedly in the discussion, and I don’t see why.

I also feel it is important to discuss previous studies that have taken a carefully controlled approach to these questions using simplified stimuli to investigate how the brain responds to stimulus timing, how responses of the human visual system underlie time perception and how stimulus features affect time perception. Again, I feel that highly relevant previous work is being ignored here. The authors should make their own choices here, but I hope they find these suggestions useful:

Bruno, A., Ayhan, I. & Johnston, A. Changes in apparent duration follow shifts in perceptual timing. J. Vis. 15, 1–18 (2015).

Heron, J. et al. Duration channels mediate human time perception. Proc. R. Soc. B Biol. Sci. 279, 690–698 (2012).

Zhou, J., Benson, N. C., Kay, K. N. & Winawer, J. Compressive temporal summation in human visual cortex. J. Neurosci. 38, 691–709 (2018).

Morrone, M. C., Ross, J. & Burr, D. Saccadic eye movements cause compression of time as well as space. Nat. Neurosci. 8, 950–954 (2005).

Johnston, A., Arnold, D. H., & Nishida, S. (2006). Spatially localized distortions of event time. Current Biology, 16(5), 472–479.

Bueti, D., Bahrami, B. & Walsh, V. Sensory and association cortex in time perception. J. Cogn. Neurosci. 20, 1054–1062 (2008).

Stigliani, A., Jeska, B. & Grill-Spector, K. Encoding model of temporal processing in human visual cortex. Proc. Natl. Acad. Sci. U. S. A. 114, E11047–E11056 (2017).

Kanai, R., Paffen, C. L., Hogendoorn, H., & Verstraten, F. A. (2006). Time dilation in dynamic visual display. Journal of Vision, 6(12):8, 1421– 1430

(And a preprint from my own lab: https://www.researchsquare.com/article/rs-579436)

Some more references about the dorsal stream and frontoparietal areas in time processing:

Hayashi, M. J., van der Zwaag, W., Bueti, D., & Kanai, R.. Representations of time in human frontoparietal cortex. Commun. Biol. 1, 1-10 (2018).

Protopapa, F. et al. Chronotopic maps in human supplementary motor area. PLoS Biol. 17, 1–34 (2019).

Harvey, B. M., Dumoulin, S. O., Fracasso, A. & Paul, J. M. A network of topographic maps in human association cortex hierarchically transforms visual timing-selective responses. Curr. Biol. 30, 1424-1434.e6 (2020).

Signed,

Ben Harvey

Reviewer #2: The paper by Sherman et al uses a previously artificial network model to predict human duration estimates of dynamic visual scenes, and then applies the same network model logic to recordings of human BOLD (fMRI) in visual areas and auditory/ somatosensory control areas. The findings replicate the previously found overestimation of scenes during which more changes occurred, both in human behavior and network output. Furthermore, BOLD activity from visual cortices shows the same bias, but only when signed changes across frames are modeled, not absolute change. BOLD from the control areas allows to reconstruct clock duration very accurately, but not the specific biases between scene types. Overall, the authors interpret their findings as evidence for time estimates being based on the neural processing of changes in visual inputs.

The approach is very interesting the results are convincing, but also quite complex. The study was preregistered during recording, and the data and codes are available.

As detailed below, some aspects need to be clarified and better explained.

1) When reading the Abstract and Introduction, I did not understand whether the authors claimed to be able to reconstruct full duration estimates, or just the biases thereof. After reading the full paper, I understand it is both, but the distinction could be made clearer from the start, especially since the authors classify the predictions of clock time as trivial in the text. (As I will detail below, I am not sure how to interpret the ability of the models to estimate clock duration so accurately.)

Some sentences in the abstract and intro use terms like 'reconstructing human subjective time estimates' (Abstract, last sentence), or 'find a one-to-one mapping' (l.15-17).

However, the studies cited in this respect (references 2–4) only test for correlations between the number of (perceived) changes and subjective duration estimates, but do not reconstruct the duration estimates (largely influenced by clock duration) themselves.

Please make this distinction and the implications thereof clearer.

2) I am surprised by the models' ability to reconstruct clock duration almost perfectly. For the human brain this is not a trivial task (providing a signal that changes over time is, but reading out a duration that correlates with clock duration from it is not).

To which extent does this linear increase in duration estimates with clock time result from the introduction of a temporal metric through the discrete time steps used in the model?

For instance, l. 174: "These ‘change’ values were standardized within-participant and compared to a criterion with

exponential decay to classify the change value as a salient event or not, giving us the

number of salient events detected by each layer for each video."

The way I understand this, the underlying discretization in time (TR) introduces clock duration into this analysis. Is this the reason for the ability to reconstruct sequence duration so well? What would be the outcome of the model (artificial and BOLD) for constant inputs or inputs with random noise?

Can you give a better explanation and rationale of these aspects, notably the exponential decay of the saliency threshold and the regression on seconds?

3) Estimation of normalized bias:

In the description (l.68), what does 'category' refer to, the duration category or the visual category (city, office)?

What is the reasoning for dividing by the mean report per category and not by the clock duration?

If the mean estimates for half of the scenes are biased (the busy city scenes), then the mean estimate also reflects this bias and the single trial bias is capturing deviations thereof.

It appears a bit uncommon to quantify biases with respect to the individual mean, rather than with respect to clock duration.

4) Visual analogue scale

Generally, it would help to add a statement that the VAS asked for estimates in seconds, which were then used in those units in further analyses. Given that the true durations were 8-24 seconds, the range depicted by the VAS is skewed. How were the ends of the scale determined? Could this have increased the underestimation of the longer durations here?

Where was the cursor placed initially?

5) Figure 2: the model-produced estimates show the same underestimation of longer durations as the human estimates. Can this be directly linked to a smaller proportion of salient events in the longer videos? How would the model estimate a concatenated series of two of the 8-s videos? As 2x8, or as ~13, as for the 16s videos?

In humans, this underestimation is attributed to regression to the mean effects, and it occurs for empty intervals, i.e. in absence of external change (Jazayeri & Shadlen 2010, Petzschner et al., 2015). How do you explain this tendency in the model? I found an explanation of this aspect in the 2019 paper and its supplementary materials, but I think it should also be explained in this work.

6) The definition of saliency is not quite clear to me.

l. 228: "The reproduction of veridical duration is trivial because, all else being

equal, longer intervals will have more salient events." Can you illustrate what would be the salient events in audition or somatosensory domain?

7) l 80: l. BFH(0,10.5) — why 10.5 ?

8) Fig.4 is lacking some detail:

Second panel: what does the color scale of the matrixes depict? What does the abbreviation TR stand for? What is X? Please spell this out in the caption.

What does the vertical black line reflect in panel 3 – zero ? Panel 4 has no axes labels.

9) p.15: Non-significant p-values do not allow to conclude on the absence of an effect. Can you provide Bayes Factors to distinguish inconclusive from null effects (see e.g. Rouder 2009), especially for the ones were the absence of an effect is important for the interpretation (control areas)?

10) l.241-244: which estimation biases are meant here? This is a bit confusing, makes it easy to overlook that the model cannot reproduce the scene-type biases.

11) Methods, p.31: please give more explanations for the decay function. Why does it vary across layers?

What would happen if there was no change, or constant random noise?

12) Fig 7., panels B, D, F: under some parameters, there seem to be significant associations between human and model bias, even for the somatosensory (low values for the upper bound), and auditory models (upper bound =2, lower bound = -2). Can you explain this?

13) Please discuss: What does it mean that the presented video duration can be predicted from visual, auditory, and somatosensory cortices? Do those regions (or the model) use the same clock?

How could a supra-modal decision be reached then? Could you speculate what would happen for videos with tones for instance?

Minor

Abstract: up to ~1 min / Methods: 8, 12, 16, 20 or 24 seconds.

p.5, l. 49: participants' subjective reports

l.79: what is the unit of Mdiff – seconds? Please add it.

Fig.3: add a color legend to the Figure

Fig.4: Panel numbers do not match the ones in the caption (3-5)

**Have the authors made all data and (if applicable) computational code underlying the findings in their manuscript fully available?**

Reviewer #1: Yes

Reviewer #2: Yes

PLOS authors have the option to publish the peer review history of their article (what does this mean?). If published, this will include your full peer review and any attached files.

Reviewer #1: **Yes: **Ben M Harvey

Reviewer #2: No
---

## [Decision Letter · Decision Letter 1]

5 Apr 2022

Dear Dr Roseboom,

Thank you very much for submitting your manuscript "Trial-by-trial predictions of subjective time from human brain activity" for consideration at PLOS Computational Biology.

As with all papers reviewed by the journal, your manuscript was reviewed by members of the editorial board and by several independent reviewers. In light of the reviews (below this email), we would like to invite the resubmission of a significantly-revised version that takes into account the reviewers' comments.

We cannot make any decision about publication until we have seen the revised manuscript and your response to the reviewers' comments. Your revised manuscript is also likely to be sent to reviewers for further evaluation.

Sincerely,

Ming Bo Cai

Associate Editor

PLOS Computational Biology

Daniele Marinazzo

Deputy Editor

PLOS Computational Biology

Reviewer's Responses to Questions

**Comments to the Authors:**

Reviewer #1: The authors have now addressed all of my comments, and I have no further concerns about this version. It's clear we take different viewpoints in places, but the neural representation of time is a new field and different researchers are welcome to have different views while we sort it all out. I thank the authors for their efforts, particularly in bringing their results into the context of vision science, by own background.

Reviewer #2: I would like to thank the authors for providing a thorough revision from which the paper has greatly benefitted.

Most of my previous points are solved (but the newly written parts have to be reread carefully for errors, see below). However, I still think that some parts of the paper can be misunderstood/ reflect an overstatement of the results: it should be made very clear that the model can estimate biases in human duration judgements, but does not provide inherent duration estimates as it relies on an externally induced metric of time. Furthermore, I think the behavioral measure needs to be better explained and justified.

1) Sentences like the following ones (and others throughout the manuscript) should be revised to coherently clearly state that the perceptual processing provides the basis for reconstructing duration estimation biases, but not the subjective time estimates as a whole. It should also be made clear why the reproduction of clock duration by the model is trivial (currently mentioned in the abstract, but statements like the one below sound contradictory).

Abstract: "Our results reveal that the information arising during perceptual processing of a dynamic environment provides a sufficient basis for reconstructing human subjective time estimates."

l. 286: "To reinforce this point, it is because all of our models – visual cortex and the control models based on auditory or somatosensory cortex - do in fact provide reasonable estimates of clock time that our key analyses focus on reproducing the subjective biases present in the reports of human participants, since it is these biases that separate clock duration from subjective duration."

2) The paragraph in the reply letter starting with "Finally, to the important point about how our models can reproduce clock time when human participants will struggle with this task…. " is very informative, and should be in the discussion of the paper.

3) The measure of bias in the analysis of the behavioral data: I am still a bit puzzled by the approach to quantify over-/ underestimation with respect to the participant's mean. Are there any references to justify this? Is the mean a meaningful parameter, i.e. are the distributions of duration reports roughly normally distributed (especially as they were collected on a VAS)?

4) Typos in the revision:

"off any events"?

Indeed, longer intervals will, on average, have a greater number

any event, task-relevant or otherwise – heart beats, eye movements, planes taking

off, etc - and so the cumulative sum of events will trivially correlate with physical

duration.

"Just as salient visual events " instead of auditory?

Just as salient auditory events would be expected to correspond to

large changes in (layer-wise) visual cortical activity, salient auditory events would be

expected to correspond to large changes in auditory cortex dynamics, and may be

trigged by, for example, hearing (or possibly imagining) a loud sound, and similarly

for somatosensory cortex.

**Have the authors made all data and (if applicable) computational code underlying the findings in their manuscript fully available?**

Reviewer #1: Yes

Reviewer #2: Yes

PLOS authors have the option to publish the peer review history of their article (what does this mean?). If published, this will include your full peer review and any attached files.

Reviewer #1: **Yes: **Ben Harvey

Reviewer #2: No
---

## [Decision Letter · Decision Letter 2]

17 May 2022

Dear Dr Roseboom,

We are pleased to inform you that your manuscript 'Trial-by-trial predictions of subjective time from human brain activity' has been provisionally accepted for publication in PLOS Computational Biology.

Best regards,

Ming Bo Cai

Associate Editor

PLOS Computational Biology

Daniele Marinazzo

Deputy Editor

PLOS Computational Biology

We would like thank the authors for your effort to address the concerns of the reviewers. We are very happy to see the work being published in PLOS Computational Biology and look forward to working with you again in the future!

Reviewer's Responses to Questions

**Comments to the Authors:**

Reviewer #2: I would like to congratulate the authors on this great paper. I have no further comments.

**Have the authors made all data and (if applicable) computational code underlying the findings in their manuscript fully available?**

Reviewer #2: Yes

PLOS authors have the option to publish the peer review history of their article (what does this mean?). If published, this will include your full peer review and any attached files.

Reviewer #2: No

---

## [Editor Report · Acceptance letter]

17 Jun 2022

PCOMPBIOL-D-21-02117R2 

Trial-by-trial predictions of subjective time from human brain activity

Dear Dr Roseboom,

I am pleased to inform you that your manuscript has been formally accepted for publication in PLOS Computational Biology. Your manuscript is now with our production department and you will be notified of the publication date in due course.

With kind regards,

Olena Szabo
